# Fabrication of the Hierarchical HZSM-5 Membrane with Tunable Mesoporosity for Catalytic Cracking of *n*-Dodecane

**Zhenheng Diao \*** , **Lushi Cheng, Xu Hou \*, Di Rong, Yanli Lu, Wenda Yue and De Sun**

School of Chemical Engineering, Advanced Institute of Materials Science, Changchun University of Technology, Changchun, Jilin 130012, China; chenglushi@outlook.com (L.C.); rongdi502@163.com (D.R.); luyanli1178@163.com (Y.L.); WendaYue2015@126.com (W.Y.); sunde2002@126.com (D.S)
**\*** Correspondence: diaozhenheng@ccut.edu.cn (Z.D.); houx@ccut.edu.cn (X.H.)

**Abstract:** Hierarchical HZSM-5 membranes were prepared on the inner wall of stainless steel tubes, using amphiphilic organosilane (TPOAC) and mesitylene (TMB) as a meso-porogen and a swelling agent, respectively. The mesoporosity of the HZSM-5 membranes were tailored via formulating the TPOAC/Tetraethylorthosilicate (TPOAC/TEOS) ratio and TMB/TPOAC ratio, in synthesis gel, and the prepared membranes were systematically characterized by X-ray diffraction (XRD), scanning electron microscope (SEM), $N_2$ adsorption–desorption, $N_2$ permeation, inductively coupled plasma (ICP), in situ fourier transform infrared (FT-IR), ammonia temperature-programmed desorption ($NH_3$-TPD), etc. It was found that the increase of the TPOAC/TEOS ratio promoted a specific surface area and diffusivity of the HZSM-5 membranes, as well as decreased acidity; the increase of the TMB/TPOAC ratios led to an enlargement of the mesopore size and diffusivity of the membranes, but with constant acid properties. The catalytic performance of the prepared HZSM-5 membranes was tested using the catalytic cracking of supercritical *n*-dodecane (500 °C, 4 MPa) as a model reaction. The hierarchical membrane with the TPOAC/TEOS ratio of 0.1 and TMB/TPOAC ratio of 2, exhibited superior catalytic performances with the highest activity of up to 13% improvement and the lowest deactivation rate (nearly a half), compared with the microporous HZSM-5 membrane, due to the benefits of suitable acidity, together with enhanced diffusivity of *n*-dodecane and cracking products.

**Keywords:** catalytic cracking; hierarchical HZSM-5 membrane; *n*-dodecane; mesopore size; diffusivity; acidity

---

## 1. Introduction

Over the last few decades, catalytic cracking of liquid hydrocarbons has been well-established for thermal management system of hypersonic aircrafts [1,2]. In this system, hydrocarbons act as, not only the power source through combustion, but also as a coolant to remove the waste heat from aircrafts, through the cracking process [3–5]. Typically, wall-coated HZSM-5 zeolite membrane could meet the demand of a hypersonic flight, where extreme working conditions, such as elevated temperature and elevated pressure (above 400 °C and 3.4–6.9 MPa, i.e., the supercritical conditions) are used, because it can diminish the pressure drop and thermal resistance of the reactor [6,7]. However, a relatively small pore size in the HZSM-5 zeolite, induces severe diffusion restrictions, slowing down the rate of hydrocarbon cracking [8–10]. Therefore, the most important issue for a better zeolite membrane is to enhance the pore-diffusion rate of the reactants.

Hierarchical zeolite membranes, combining the advantages of both micro- and meso-porous zeolites, have received a significant attention during recent years, due to its improved catalytic

performance and fluid permeability [11,12]. Generally, hierarchical zeolite powder is synthesized, first, and then used for preparing the hierarchical zeolite membrane through the wash-coating method. Zhang et al. prepared hierarchical coating membranes with desilication-treated HZSM-5 zeolites, and found that the catalytic cracking performance was significantly improved, due to the enhanced diffusion properties of the coatings [13]. Ji et al. wash-coated the MFI nanosheet zeolites on the inner surface of the stainless steel tubes, for the catalytic *n*-dodecane cracking, and found that the mesopores in the MFI zeolite, benefited the catalytic performance [14]. Similarly, we also synthesized wall-coated meso-HZSM-5@MCM-41 zeolites, and found that the hierarchical structure facilitated acid site accessibility and, thus, improved the catalytic cracking activity [15]. However, the mechanical stability of the coating membrane was unsatisfactory. Liu et al. coated the HZSM-5 zeolite on the inner wall of the stainless steel tube and found that the mass loss of the coatings, after an ultrasonic treatment, reached up to 12.5% [16]. Furthermore, the binder ($SiO_2$) in wall-coated catalysts is inactive, which would lower the overall activity of coatings.

Hydrothermal growth, with stronger adhesions of zeolite membranes, to support a controllable growth rate without binders, was used for preparing the hierarchical zeolite membrane, alternatively. Richter et al. [17] and Wang et al. [18] created additional non-zeolite pores in the inter-crystalline, grain boundaries of the ZSM-5 membrane, using alcohol in the synthesis gel, and got a remarkable enhancement of gas permeances. However, it should be noted that the diameters of the crystalline grains were above 2 μm, in their studies, thus, the intracrystal diffusion limitation was still a problem that needed to be improved. Liu et al. prepared a hierarchically porous TS-1 zeolite membrane, through a secondary growth method, using desilicated TS-1 powder as seed, and the hierarchical membrane exhibited a superior $CO_2/N_2$ separation properties, in comparison to the conventional one [19]. Wang et al. created mesopores in the HZSM-5 membrane with NaOH-treatment and then employed the membrane for catalytic cracking of supercritical *n*-dodecane. The results showed that the intracrystal mesopores in the membrane, facilitated the cracking activity and stability [20]. However, the desilication method used, faced a significant loss of acid sites originating at the expense of microporosity and a poor control over mesoporosity [21]. Alternatively, amphiphilic organosilanes were employed as a meso-porogen for synthesis of hierarchical zeolites. Due to the self-assembly of the amphiphilic organosilanes, in the synthesis gel, the uniform and ordered mesopores, with size of ~3 nm, is introduced into the zeolite crystals, as a result [22,23]. Peng et al. fabricated the hierarchical HZSM-5 membrane on the porous $Al_2O_3$ hollow fibers, with TPOAC as a meso-porogen, and investigated the $TPOAC/SiO_2$ ratio, synthesis temperature, and crystallization time, on the microstructures and filtration performance. It was found that the suitable $TPOAC/SiO_2$ ratio was of 0.02–0.03, and only nanoparticles were formed rather than membranes, when the ratio was above the suitable value [24]. As is known, it is insufficient for the $TPOAC/SiO_2$ ratio to significantly increase its mesoporosity, when the ratio value is below 0.03 [25–27]. The catalytic cracking performance was closely related to the mesoporosity of the zeolites [28–30]. Therefore, an important concern for further improving the catalytic cracking performances of zeolite membranes, is to prepare a membrane with a high mesoporosity.

In this work, we put focus on the synthesis of hierarchical HZSM-5 membranes, with a high mesoporosity, as well as the effects of synthesis recipes on the mesoporosity, acidity, diffusivity, and catalytic cracking performance of membranes. First, hierarchical HZSM-5 membranes were prepared on the inner surface of stainless steel tubes. The mesoporosity of the membrane was adjusted by the TPOAC/TEOS and the TMB/TPOAC ratios, in the synthesis gel. Then, the texture properties, diffusivity, and acidities of the synthesized membranes were characterized by various instrumental methods. Finally, these membranes were employed as catalysts for catalytic cracking of the supercritical *n*-dodecane.

## 2. Results and Discussion

### 2.1. Structures Characterization

Figure 1 displays the XRD patterns of the HZSM-5 membranes. For all samples, there were several typical diffraction peaks corresponding to the characteristic peaks of the MFI topology, indicating a successful crystallization of the HZSM-5 phase on the stainless steel tubes (SST) [18]. With the TPOAC/TEOS ratios in the synthesis gel growing from 0 to 0.15, the relative crystallinities of the HZSM-5 membranes decreased gradually (as shown in Table 1), which was in line with the observation of Peng et al. [24] and Cho et al. [31]. The peak intensities reduced slightly with an increase in the TMB/TPOAC ratios, indicating that the TMB had a negligible effect on the HZSM-5 crystallization. Additionally, the intensity of the stainless steel support signals ($2\theta = 43.8°$) for the ZM-0.15TP was significantly lower than that for the ZM-0TP, which might be attributed to the increased thickness of the zeolite membranes on the stainless steel tubes. Wang et al. also observed that the increase in thickness of the zeolite membranes would reduce the intensity of the stainless steel support signals [20].

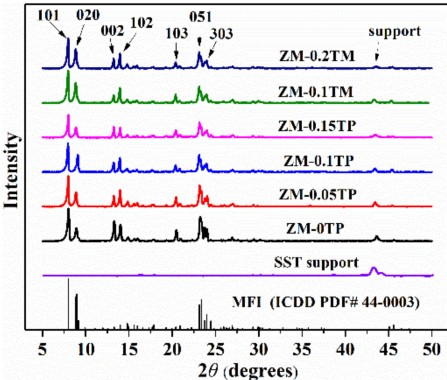

**Figure 1.** XRD patterns of the HZSM-5 membranes on the stainless steel tube.

**Table 1.** Physical properties and catalytic performances of the zeolite membranes.

| Membranes | $R_c$ [a] (%) | $\delta$ [b] (μm) | Loading amounts (mg·cm$^{-2}$) | Density [c] (g·cm$^{-3}$) | N$_2$ permeance mol·Pa$^{-1}$·s$^{-1}$·m$^{-2}$ | $r_d$ [d] (%) |
|---|---|---|---|---|---|---|
| ZM-0TP | 100 | 5.08 ± 0.25 | 1.40 ± 0.07 | 2.76 | 2.7×10$^{-6}$ | 36.79 |
| ZM-0.05TP | 87 | 5.27 ± 0.24 | 1.37 ± 0.06 | 2.60 | 4.1×10$^{-5}$ | 35.17 |
| ZM-0.1TP | 71 | 5.43 ± 0.31 | 1.35 ± 0.09 | 2.49 | 1.6×10$^{-4}$ | 32.72 |
| ZM-0.15TP | 61 | 6.14 ± 0.27 | 1.31 ± 0.05 | 2.13 | 5.4×10$^{-3}$ | 16.35 |
| ZM-0.1TM | 68 | 5.59 ± 0.32 | 1.41 ± 0.06 | 2.52 | 2.2×10$^{-4}$ | 27.26 |
| ZM-0.2TM | 68 | 5.72 ± 0.24 | 1.34 ± 0.10 | 2.34 | 8.5×10$^{-4}$ | 20.10 |

[a] $R_c$: XRD relative crystallinity. [b] $\delta$: mean thickness determined by SEM. [c] Density calculated from the mass loading and mean thickness. [d] Deactivation rate ($r_d$) is defined as: $r_d = (X_{t=5} - X_{t=30})/X_{t=5} \times 100\%$, wherein $X_{t=5}$ and $X_{t=30}$ are the *n*-dodecane conversions, on steam, at 5 and 30 min, respectively.

The top-view SEM images (Figure 2) showed that the crystals of the ZM-0TP exhibited a typical coffin-shape and epitaxially grew into a continuous HZSM-5 membrane. With increasing TPOAC/TEOS ratios, the crystal size of the HZSM-5 zeolites got smaller, and the top surface of the HZSM-5 membrane became rougher and looser, which might be attributed to the promotion of the mesopores and interruption of the crystal growth caused by the TPOAC [24]. For the ZM-0.1TM and ZM-0.2TM, the HZSM-5 membranes were as continuous as that for the ZM-0.1TP, but the roughness of the crystal surface was obviously higher than that for the ZM-0.1TP. This was probably because the mesopores in the membranes had got swollen because of the TMB [32].

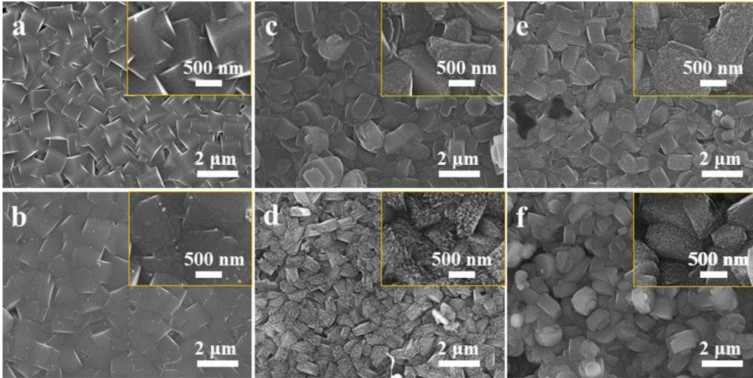

**Figure 2.** Top-view SEM images of the zeolite membranes. (**a**) ZM-0TP, (**b**) ZM-0.05TP, (**c**) ZM-0.1T, (**d**) ZM-0.15TP, (**e**) ZM-0.1TM, and (**f**) ZM-0.2TM.

Figure 3 displays the cross-view SEM images of the zeolite membranes. It can be seen that the membrane thickness increased, gradually, with both the TPOAC/TEOS and TMB/TPOAC ratios. This was consistent with the XRD results and demonstrated that a decrease in the XRD signals of the stainless steel support was the result of an increased thickness of the zeolite membranes. Table 1 presents the thickness, mass loading, and density of the zeolite membranes. For all membranes, the mass loading was similar. Thus, a decreased trend for membrane density was observed with increasing both the TPOAC/TEOS and the TMB/TPOAC ratios. Compared to the observations of Li et al. [24], a continuous HZSM-5 membrane was successfully prepared in this work, although the TPOAC/TEOS ratio was of 0.15 (as shown in Figures 2d and 3d). This would be attributed to the treatment of the seeded support with a TPOAC solution, which was in favor of a strong interaction between the seeded support and the silica species.

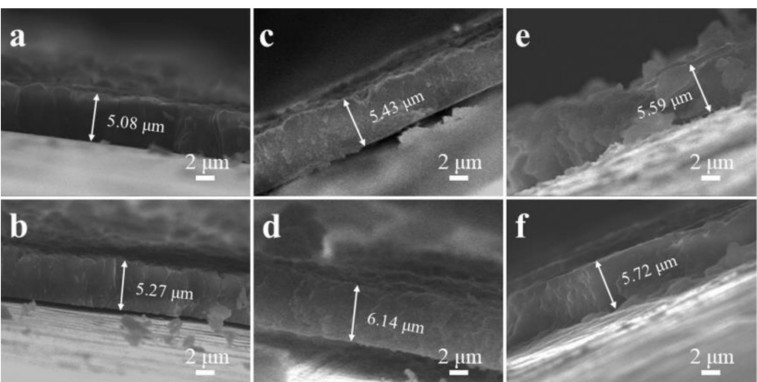

**Figure 3.** Cross-view SEM images of the zeolite membranes. (**a**) ZM-0TP, (**b**) ZM-0.05TP, (**c**) ZM-0.1TP, (**d**) ZM-0.15TP, (**e**) ZM-0.1TM, and (**f**) ZM-0.2TM.

Nitrogen adsorption isotherms and pore size distribution curves of the zeolite powders are illustrated in Figure 4. As shown in Figure 4a, the ZM-0TP showed a nitrogen adsorption isotherm of type-I, which is typical for microporous zeolites, according to the International Union of Pure and Applied Chemistry (IUPAC) classification. The others displayed type IV with a capillary condensation loop at a relative pressure, higher than 0.4, which indicated that the samples possessed significant mesoporous features. The mesopore size distribution curve (Figure 4b) of the ZM-0TP reflected undetectable mesopores in the range of 2–10 nm, while those of ZM-0.05TP, ZM-0.1TP, and ZM-0.15TP exhibited narrow pore size distribution curves at around 3.1 nm, demonstrating the presence of uniform mesopores. Furthermore, the curve intensity increased significantly from ZM-0.05TP to ZM-0.15TP, indicating the amount of uniform mesopores increased with the TPOAC/TEOS ratios. With the increase of the TMB/TPOAC ratios, the mesopore size had expanded to 3.8 nm for the ZM-0.1TM,

and further to 4.3 nm for the ZM-0.2TM. This demonstrated that the difference in membrane roughness of the SEM results, was due to the variation of membrane mesoporosity.

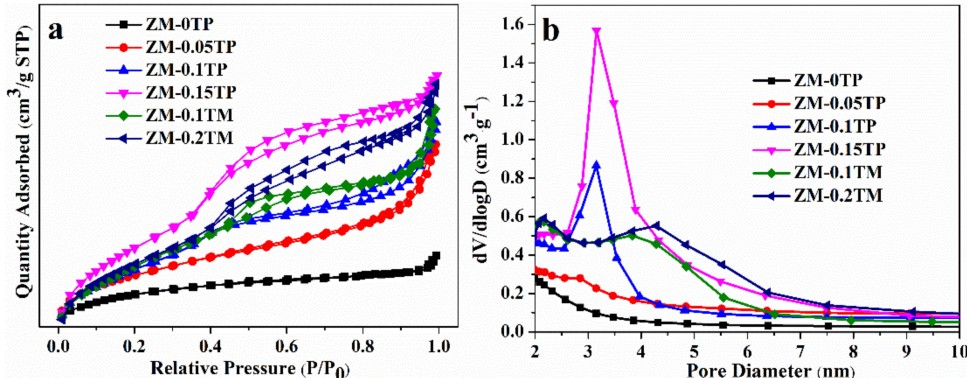

**Figure 4.** Porous properties of the zeolites. (**a**) $N_2$ adsorption–desorption isotherms, (**b**) Barrett–Joyner–Halenda (BJH) adsorption pore size distribution.

The values of the total specific surface area ($S_{BET}$), total volume ($V_t$), external specific area ($S_{exter}$), and the microporous volume ($V_{micro}$) were calculated; summarized in Table 2. $V_{micro}$ gradually decreased from 0.133 cm$^3$·g$^{-1}$ for the ZM-0TP to 0.071 cm$^3$·g$^{-1}$ for the ZM-0.15TP, which was in accordance with the downtrend of their XRD crystallinities. Both, $S_{BET}$ and $V_t$ of the zeolites, increased with the TPOAC/TEOS and the TMB/TPOAC ratios, which could be ascribed to the enlargement of the mesopore amount and mesopore size, respectively.

**Table 2.** Textural and acid properties of the zeolite powders.

| Samples | $S_{BET}$ (m$^2$·g$^{-1}$) | $S_{exter}$ (m$^2$·g$^{-1}$) | $V_t$ (cm$^3$·g$^{-1}$) | $V_{micro}$ (cm$^3$·g$^{-1}$) | Si/Al [a] | Acid density [b] ($\mu$mol NH$_3$·g$^{-1}$) | | | | | | |
|---|---|---|---|---|---|---|---|---|---|---|---|---|
| | | | | | | $B_W$ | $B_S$ | $L_W$ | $L_S$ | T | $T_W$ | $T_S$ |
| ZM-0TP | 432 | 146 | 0.231 | 0.133 | 71 | 86 | 82 | 46 | 37 | 251 | 132 | 119 |
| ZM-0.05TP | 491 | 218 | 0.353 | 0.126 | 86 | 47 | 43 | 78 | 66 | 234 | 125 | 109 |
| ZM-0.1TP | 515 | 316 | 0.384 | 0.091 | 90 | 40 | 39 | 69 | 63 | 211 | 109 | 102 |
| ZM-0.15TP | 586 | 425 | 0.453 | 0.071 | 107 | 15 | 10 | 88 | 35 | 148 | 103 | 45 |
| ZM-0.1TM | 528 | 344 | 0.390 | 0.083 | 92 | 38 | 36 | 72 | 66 | 212 | 110 | 102 |
| ZM-0.2TM | 533 | 349 | 0.433 | 0.085 | 93 | 41 | 37 | 67 | 62 | 207 | 108 | 99 |

[a] Si/Al ratio from the ICP measurement. [b] Acid density evaluated by combing the NH$_3$-TPD and the Py-FTIR results. B, L, T, S, and W represent the Brønsted, Lewis, total, strong, and weak acid site, respectively.

## 2.2. Gas Permeation Properties

$N_2$ permeance through the zeolite membranes was selected to reflect the diffusivity of the membrane; it has been reported that the high permeance value implies a high diffusivity [18,33]. As seen in Table 1, the values of the $N_2$ permeance increased, monotonously, with both the TPOAC/TEOS and TMB/TPOAC ratios. For example, the $N_2$ permeance through the ZM-0TP was about $2.7 \times 10^{-6}$ mol·Pa$^{-1}$·s$^{-1}$·m$^{-2}$, while it was $1.6 \times 10^{-4}$ mol·Pa$^{-1}$·s$^{-1}$·m$^{-2}$ through the ZM-0.1TP and $5.4 \times 10^{-3}$ mol·Pa$^{-1}$·s$^{-1}$·m$^{-2}$ through the ZM-0.15TP, which was consistent with the increasing trend of the mesopore amount. As for the ZM-0.2TM, the $N_2$ permeance was $8.5 \times 10^{-4}$ mol·Pa$^{-1}$·s$^{-1}$·m$^{-2}$, which was 5.3 times higher than that for the ZM-0.1TP. This implied that the diffusivity of the zeolite membrane could be greatly improved by increasing, both, the mesopore amount and the mesopore size in membranes. Ji et al. measured the $N_2$ gas permeation of the b-oriented bi-layered HZSM-5 membranes, and found that the membrane with a thickness of 0.8 μm showed a permeation of $6.3 \times 10^{-5}$ mol·Pa$^{-1}$·s$^{-1}$·m$^{-2}$ [33], which was significantly lower than that of the ZM-0.2TM. This further demonstrated that the mesopores in the membrane benefited the diffusivity.

## 2.3. Compositions and Acid Properties

The Si/Al ratios of the zeolite powders were measured by the ICP; the results are shown in Table 2. As can be seen, the Si/Al ratios increased with the TPOAC/TEOS ratios, indicating that TPOAC had a negative effect on the incorporation of Al into the zeolites. For the ZM-0.1TP, the ZM-0.1TM, and the ZM-0.2TM, the Si/Al ratios were similar to each other.

Figure 5 presents the $NH_3$-TPD measurements of the zeolite powders. For all samples, there were two peaks at around 175–180 °C and 360–380 °C, corresponding to the weak and strong acid sites, respectively. The peak intensities of, both, weak and strong acid sites decreased by the following order: ZM-0TP > ZM-0.05TP > ZM-0.1TP > ZM-0.15TP, while the maximum temperature of the peaks was similar. This suggested that the acid amount decreased and acid strength remained unchanged, with increasing TPOAC/TEOS ratios. The acidity of the zeolite powder was further characterized by the Py-FTIR (as shown in Figure 6). All zeolites showed characteristic bands at around 1452, 1542, and 1489 cm$^{-1}$, which could be assigned to the pyridine interacting with the Lewis, Brønsted, and both acid sites [34,35]. By combining the results of the $NH_3$-TPD and pyridine-adsorbed FT-IR, acid density and strength were evaluated and are listed in Table 2. The amount of, both, total acid sites and strong B acid sites decreased with increasing the TPOAC/TEOS ratios. Especially for the ZM-0.15TP, the amount of strong B acid was only 10 μmol $NH_3 \cdot g^{-1}$, which was about one-eighth of that for ZM-0TP and one-fourth of that for the ZM-0.1TP. This could be ascribed to the enlargement of the Si/Al ratios of the zeolites. Qu et al. investigated the relationship between the Si/Al ratios and the acid properties of the HZSM-5 zeolite, and found the decrease of the Si/Al ratios enhanced the amount of, both, total and strong B acid sites [36]. However, that of the weak L acid sites exhibited a contradictory trend. It was probably because higher proportions of Al were located at the extra-framework of the zeolites, with increasing TPOAC/TEOS ratios. Koekkoek et al. prepared hierarchical HZSM-5 zeolites with TPOAC as the meso-porogen, and observed that the proportions of extra-framework Al in the hierarchical HZSM-5 zeolite were significantly higher than that in the conventional one [37]. In addition, the ZM-0.1TM and the ZM-0.2TM showed a similar acid type and amount for the ZM-0.1TP, indicating that the TMB had a negligible effect on the acidity of zeolites.

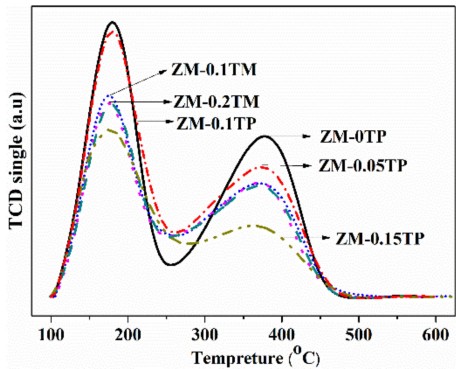

**Figure 5.** $NH_3$-TPD profiles of the zeolite powders.

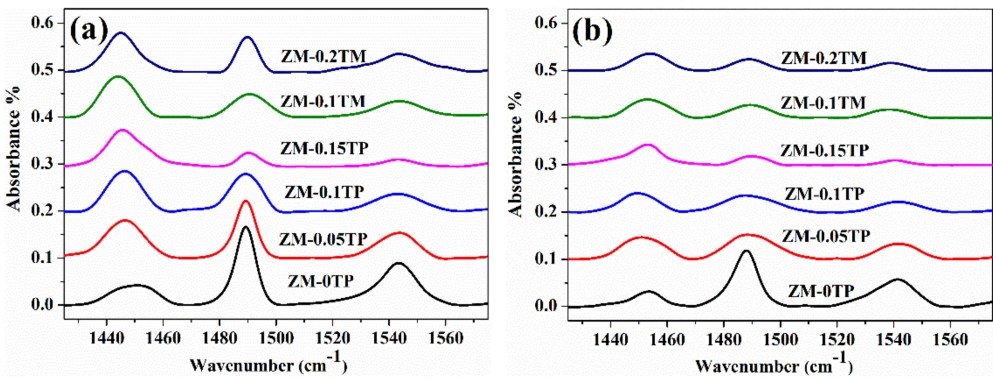

**Figure 6.** Pyridine-adsorbed FT-IR spectra of the zeolite powders at (**a**) 150 °C and (**b**) 350 °C.

## 2.4. Catalytic Cracking Performances

### 2.4.1. Catalytic Cracking Activities

Catalytic cracking of the supercritical *n*-dodecane over the zeolite membranes were carried out in a flowing reactor, at 500 °C and 4 MPa, and Figure 7 depicted the *n*-dodecane conversion as a function of the time on stream (TOS). For the ZM-0TP, the initial *n*-dodecane conversion was 19.26%, which sharply dropped to 12.17% at TOS = 30 min, with a deactivation rate ($r_d$) of 36.79% (as shown in Table 1). With increasing TPOAC/TEOS ratios, the initial *n*-dodecane conversion increased to 25.50% for the ZM-0.05TP and a further to 29.07%, for the ZM-0.1TP, which would be the result of the improved diffusivity. On the contrary, theZM-0.15TP showed an opposite result (with an initial conversion of 18.33%), which might be ascribed to the lowest amount of strong and B acid sites, although it exhibited the best diffusivity. The $r_d$ decreased, monotonously, with increasing the TPOAC/TEOS ratios, suggesting that both the enhanced diffusivity and decreased amount of acid sites improved the catalytic stability of the membranes. ZM-0.1TM and ZM-0.2TM showed a higher catalytic activity than the ZM-0.1TP, although their acid properties were similar, because the larger mesopore size benefited the diffusion of the reactant and, thus, facilitated the catalytic properties.

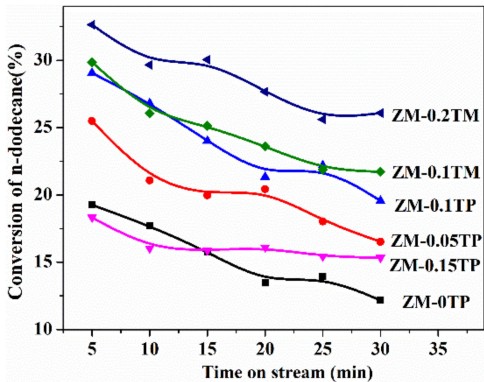

**Figure 7.** Catalytic conversion of the supercritical *n*-dodecane cracking over the zeolite membranes.

### 2.4.2. Product Distribution

Table 3 revealed the difference in the average conversion and product distribution of *n*-dodecane cracking over the zeolite membranes. It is well-known that hydrogen transfer for olefin consuming and dehydrogenation–cyclization for aromatics formation are, both, typical secondary reactions during the process of hydrocarbon cracking, which are likely to depend on the diffusivity of products, as well as on the acid site density [38]. The high ratios olefin/paraffin (o/p) in the C3 and C4 products, especially for the isobutene/isobutane, implies a low extent of the hydrogen transfer reactions [38,39]. It can be seen in Table 3 that the $C3^=/C3$ and $i\text{-}C4^=/i\text{-}C4$ ratios increased in the order ZM-0.15TP >

ZM-0.1TP > ZM-0.05TP > ZM-0TP, which was in good agreement with the catalytic stability of the membranes. This would be the result of, both, the high diffusivity of the membrane and a low acidity. For the ZM-0.1TP, the ZM-0.1TM, and the ZM-0.2TM, the o/p ratios increased with the mesopore size, due to an improved diffusivity of membranes, in line with the catalytic activities and stabilities. When looking into the selectivities of aromatics (benzene and toluene), it was obvious that the selectivities of aromatics were opposite to the o/p ratios. This further demonstrated that, both, the high diffusivity of the membrane and the low acidity, limited a secondary reaction and, thus, improved the stability of zeolites.

**Table 3.** Average conversion and mass selectivity of the *n*-dodecane cracking over the zeolite membranes.

| Products | ZM-0TP | ZM-0.05TP | ZM-0.1TP | ZM-0.15TP | ZM-0.1TM | ZM-0.2TM |
|---|---|---|---|---|---|---|
| methane | 4.03 | 3.58 | 3.34 | 2.37 | 2.97 | 2.45 |
| ethylene | 4.79 | 5.42 | 5.77 | 4.89 | 6.43 | 6.31 |
| ethane | 4.66 | 4.37 | 5.04 | 3.58 | 5.87 | 5.62 |
| propane | 5.42 | 6.01 | 7.45 | 4.92 | 7.04 | 7.33 |
| propylene | 9.11 | 10.45 | 13.26 | 11.07 | 12.76 | 13.7 |
| iso-butane | 0.59 | 0.46 | 0.39 | 0.24 | 0.37 | 0.31 |
| *n*-butane | 3.35 | 2.39 | 2.01 | 1.43 | 2.10 | 2.05 |
| *trans*-2-butene | 1.38 | 1.17 | 0.92 | 0.87 | 1.08 | 1.74 |
| iso-butene | 1.46 | 1.41 | 1.57 | 1.35 | 1.69 | 1.66 |
| 1-butene | 4.07 | 2.98 | 2.51 | 2.85 | 2.56 | 3.29 |
| *cis*-2-butene | 1.21 | 0.85 | 1.39 | 0.58 | 0.74 | 0.85 |
| pentene | 6.72 | 7.37 | 6.24 | 5.49 | 6.38 | 6.62 |
| *n*-pentane | 5.95 | 6.19 | 5.76 | 3.32 | 5.85 | 4.54 |
| hexene | 4.74 | 5.83 | 5.85 | 6.31 | 5.26 | 5.93 |
| *n*-hexane | 2.57 | 5.09 | 4.13 | 3.25 | 3.87 | 3.71 |
| benzene | 0.24 | 0.21 | 0.18 | 0.07 | 0.13 | 0.09 |
| heptene | 4.58 | 5.56 | 5.57 | 7.02 | 6.09 | 5.84 |
| *n*-heptane | 4.03 | 2.83 | 3.11 | 3.59 | 2.78 | 2.47 |
| toluene | 0.41 | 0.35 | 0.30 | 0.17 | 0.32 | 0.21 |
| octene | 5.37 | 3.15 | 6.85 | 7.45 | 6.34 | 6.67 |
| *n*-octane | 3.41 | 2.06 | 1.36 | 2.73 | 1.38 | 1.52 |
| nonene | 5.27 | 3.97 | 4.88 | 7.16 | 5.97 | 5.06 |
| *n*-nonane | 4.16 | 3.68 | 2.56 | 3.23 | 2.04 | 2.24 |
| decene | 4.79 | 5.97 | 3.92 | 6.86 | 4.16 | 4.49 |
| *n*-decane | 2.64 | 2.33 | 1.19 | 1.75 | 1.06 | 0.85 |
| undecene | 1.80 | 1.78 | 1.54 | 3.64 | 1.73 | 1.86 |
| *n*-undecane | 2.94 | 3.89 | 2.44 | 2.87 | 2.62 | 2.07 |
| dodecene | 0.31 | 0.65 | 0.47 | 0.94 | 0.41 | 0.52 |
| C3$^=$/C3 | 1.68 | 1.74 | 1.78 | 2.25 | 1.81 | 1.87 |
| i-C4$^=$/i-C4 | 2.47 | 3.07 | 4.03 | 5.63 | 4.57 | 5.35 |
| Average conversion (%) | 15.39 | 20.26 | 23.84 | 16.18 | 24.71 | 28.62 |

Among all membrane catalysts, the ZM-0.2TM exhibited the highest average activity (28.62%), with an above 13% improvement, compared to the ZM-0TP (as shown in Table 3). Furthermore, an excellent catalytic stability and limited secondary reaction was also observed over the ZM-0.2TM, with an $r_d$ of 20.10% (as shown in Table 1), which was nearly a half of that of the ZM-0TP. Studies on the catalytic cracking revealed that the catalytic activities of the hierarchical HZSM-5 membranes varied, monotonously, neither with diffusivity nor with the amount of acid sites, demonstrating that the excellent catalytic performance of ZM-0.2TM could be ascribed to the benefits of, both, a suitable acidity and an enhanced diffusion of the *n*-dodecane and the cracking products. Compared with the studies mentioned above, the ZM-0.2TM also showed an excellent catalytic performance. For instance, the desilicated HZSM-5 membrane prepared by Wang et al. showed a lower Si/Al ratio and a higher loading amount [20]. However, its average conversion of supercritical *n*-dodecane cracking was of ca. 20%, which was distinctly lower than the ZM-0.2TM. Similarly, the MFI nanosheet coatings synthesized

by Ji et al. also exhibited a lower Si/Al ratio, a higher loading amount and a lower catalytic activity at 500 °C, for the supercritical *n*-dodecane cracking [14], although the flow-rate of *n*-dodecane used in the literature was only one-half of that in this paper. Ji et al. investigated the catalytic cracking of the supercritical *n*-dodecane, over the bi-layered HZSM-5 membranes [33]. It was found that the b-oriented, bi-layered membrane exhibited a lower catalytic activity, due to the less loading amount and acid sites. However, both, the ratios of $C3^=/C3$ and $i\text{-}C4^=/i\text{-}C4$ were also lower than that of the ZM-0.2TM. This would be the result of the superior diffusion properties of the ZM-0.2TM, which thus, improved the catalytic activity and limited a secondary reaction.

## 3. Experiment

### 3.1. Materials

[3-(trimethoxysilyl)propyl]octadecyldimethylammonium chloride (TPOAC, 60 wt % methanol solution), Tetrapropylammonium hydroxide (TPAOH, 25 wt % in water), and mesitylene (TMB, 99%) were purchased from J&K Chemical Ltd. (Beijing, China). Tetraethylorthosilicate (TEOS) and aluminum nitrate [$Al(NO_3)_3 \cdot 9H_2O$] were purchased from the Guangfu Chemical Reagent Company (Tianjin, China). *n*-Dodecane with 99.5% purity was purchased from the Sinopharm Chemical Reagent Co., Ltd. (Shanghai, China). All chemicals were used as received, without further purification.

Non-porous 304 stainless-steel tubes ((SST), 300 mm in length, 2 mm id) and porous stainless steel slabs ((PSSS), 30 mm in diameter, mean pore size of 250 nm) were used as substrates. Prior to use, both PSSS and SST were washed with acetone and distilled water, to remove adsorbed impurities, and then dried at 120 °C, overnight.

### 3.2. Preparation of the Seed Layers

Silicate-1 seed solution with an average crystal size of 100 nm was prepared, with a molar composition of 0.36 TPAOH:1 TEOS:58 $H_2O$. The synthesis gel was placed in a Teflon-lined, stainless-steel autoclave and crystallized at 100 °C, for 12 h. Then the as-synthesized seed solution was filled into the stainless steel tube, and the tube was sealed and ultrasonic treated (55 kW, 40 kHz), for 30 min. The residual solution was poured, and the tube was dried at 100 °C, overnight, and calcined at 450 °C, for 2 h. The mass loading of seeds, per tube, was 2.2 ± 0.1 mg for all tubes. After that, the tube was treated with the TPOAC solution (0.5 wt %) by the wash-coating method and dried at 60 °C, overnight.

### 3.3. Preparation of the Zeolite Membranes

Zeolite membranes were prepared on the seeded SST, by in-situ hydrothermal crystallization. TEOS, TPAOH, $Al(NO_3)_3$, and $H_2O$ were well-mixed, in a glass bottle, and then a measured amount of the TPOAC and the TMB was added. The mixture with a molar composition of 0.32 TPAOH:1 TEOS:165 $H_2O$:0.01 $Al(NO_3)_3$:$x$ TPOAC:$y$ TMB was aged for 24 h, at ambient temperature. Then, the SST was filled with the synthesis gel and placed vertically, in an oven, at 165 °C, for 48 h. After that, the SST was washed with water and followed by treatment in an ultrasound bath, for 15 min, to remove the loosely attached crystals on the membrane surface. Finally, the SST was dried at 100 °C, overnight, and calcined at 450 °C, for 2 h. The obtained samples were denoted as ZM-0TP, ZM-0.05TP, ZM-0.1TP, ZM-0.15TP, ZM-0.1TM, and ZM-0.2TM, corresponding to $x = 0$, $y = 0$; $x = 0.05$, $y = 0$; $x = 0.1$, $y = 0$; $x = 0.15$, $y = 0$; $x = 0.1$, $y = 0.1$; $x = 0.1$, $y = 0.2$; respectively.

Zeolite membranes were also prepared on the PSSS with similar preparation procedures, conditions, and recipes to that of the SST, except for the seeded ones and for the crystallization holder. Seeded PSSS was prepared in a 50 mL beaker containing the seed solution and the HZSM-5-coated PSSS was synthesized in a 50 mL Teflon-lined, stainless steel autoclave, with a seeded PSSS placed, vertically, in the synthesis solution. The as-prepared membranes on the PSSS were used only for nitrogen permeation tests.

The zeolite powders were obtained by recovering the solids formed, simultaneously, in the synthesis solutions with the zeolite membranes, and were further treated at the same conditions as that of the membranes. The powder samples were used for textural and acid characterization.

### 3.4. Characterization

The XRD measurements were carried out on a Rigaku D-max2500 V/PC X-ray diffractometer (Rigaku Corporation, Tokyo, Japan), using Cu-Kα radiation source. XRD relative crystallinity of the zeolite membrane was calculated, based on the peak intensity at $2\theta = 23.1°$, as compared to that of the ZM-0TP [40]. Scanning electron microscopy (SEM) was obtained using a JEOL7610 scanning electron microscope (FE-SEM). $N_2$ adsorption–desorption processes were conducted using a nanopore analyzer (ASAP-2020), at 77 K. The Brunauer-Emmett-Teller (BET) method and the t-plot method were applied to calculate the total specific surface area ($S_{BET}$) [41], the volume of the micropores ($V_{micro}$), and the external surface area ($S_{exter}$) [42], respectively. $N_2$ uptake at a relative pressure ($P/P_0$) of 0.99 and the BJH model, following the adsorption branch of the isotherm were used for evaluation of the total pore volume ($V_t$) [43], and the mesopore size distribution [44], respectively. The Si/Al ratios of the zeolite powder were collected by an ICP-900 (N + M) inductive-coupled plasma emission spectrometer.

$NH_3$ temperature-programmed desorption was measured on a Quantachrome CHEMBET 3000 (Quantachrome Instruments, Boynton Beach, FL, USA). First, the catalyst of 0.05 g was pretreated at 600 °C and then cooled to 50 °C. Second, the catalyst was saturated by ammonia (20% $NH_3$ in He). Finally, physically adsorbed $NH_3$ was removed at 100 °C, for 60 min, and the temperature was raised to 600 °C, with a heating rate of 10 °C /min, to measure the amount of chemically adsorbed $NH_3$. Pyridine FTIR spectra was collected on a FT-IR Bruker Equinox spectrometer (Vertex 70, Billerica, MA, USA). After being pretreated at 400 °C, for 1h, under a residual pressure below $10^{-4}$ Pa, the samples were saturated by pyridine. Then the spectra were collected at 150 °C and 350 °C, which was assigned to the total and strong acid sites, respectively.

### 3.5. $N_2$ Gas Permeation Test

$N_2$ gas permeation was measured at an ambient temperature, using the HZSM-5 membranes grown on the PSSS, according to Wang et al. [18]. The upstream pressure was controlled by a backpressure regulator, whereas, the downstream gas was vented to air. The permeation rates of the $N_2$ gas were measured by a bubble flowmeter.

### 3.6. Catalytic Cracking of n-dodecane

The catalytic cracking of the supercritical *n*-dodecane was carried out to evaluate the catalytic performance of the zeolite membranes, as described in our previous work [15]. *n*-Dodecane was pumped into the reactor by a plunger pump with a flow rate of 10 mL/min. The temperature of the reaction tube (500 °C) was monitored by a thermocouple, and controlled by a proportional–integral–derivative (PID) controller. The pressure was controlled by a Tescom back-pressure valve and fixed at 4 MPa. The gaseous and liquid samples were cooled, separated, and collected at an interval of 5 min.

The liquid samples were analyzed using an Agilent 7890B GC, with a flame ionization detector and a HP-5 column (30 m × 0.32 mm). The gaseous samples collected were analyzed on a GC (Agilent 6890, Agilent, Santa Clara, CA, USA) equipped with a flame ionization detector and a HP-PLOT/Q column (30 m × 0.53 mm). The catalytic conversion was defined as the ratio of the consumed *n*-dodecane to the fed *n*-dodecane.

## 4. Conclusions

Hierarchical HZSM-5 zeolite membranes, with uniform mesopores were prepared using TPOAC as the mesoscale template and TMB as the swelling agent. The ratios of, both, the TPOAC/TEOS and the TMB/TPOAC, in the synthesis gel, showed an important influence on the mesoporosity,

acidity, and catalytic performance of the hierarchical HZSM-5 membranes. The increase of the TPOAC/TEOS ratios induced an enlargement of specific surfaces area of zeolites and diffusivity of the membranes, together with a decrease of, both, Brønsted and strong acid site amounts. The increase of the TMB/TPOAC ratios led to an increase of the mesopore size and diffusivity, but with constant acid properties. The hierarchical HZSM-5 membrane prepared with the TPOAC/TEOS ratio of 0.1 and TMB/TPOAC ratio of 2, exhibited a high conversion of *n*-dodecane cracking and stability, as well as a low degree of secondary reactions, ascribing to the benefits of suitable acidity, together with an enhanced diffusivity of membrane.

**Author Contributions:** All authors conceived and designed the experiments; Z.D., L.C., D.R., and Y.L. performed the experiments; X.H., W.Y., and D.S. analyzed the data; and Z.D. wrote the paper.

**Funding:** This work was financially supported by the National Key R&D Plan under the Grant Nos. 2018YFB0504600 and 2018YFB0504603.

**Conflicts of Interest:** The authors declare no conflict of interest.

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
