# Peer review of "Fabrication of the Hierarchical HZSM-5 Membrane with Tunable Mesoporosity for Catalytic Cracking of n-Dodecane"

_catalysts, doi:10.3390/catal9020155_

Round 1

Reviewer 1 Report

The comparison with the literature data have been introduced and discussed in the manuscript as suggested. The manuscript could be now accepted for publication.

Author Response

Point 1: The comparison with the literature data have been introduced and discussed in the manuscript as suggested. The manuscript could be now accepted for publication.

Response 1: Thanks for your comment.

Reviewer 2 Report

The authors have addressed most of my concerns in the last round of review and revision. This work can be considered as a publication in Catalysts. But before that, I still have a minor question about the XRD (Figure 1). Although they added the standard diffraction peaks of HZSM-5 (PDF#44-0003), I am confused why these standard peaks don't match well with those of HZSM-5 membranes with respect to the peak positions, quantity, etc. Please double check this and revise the manuscript accordingly. Suggestion is a minor revision.

Author Response

This manuscript is a resubmission of an earlier submission. The following is a list of the peer review reports and author responses from that submission.

Round 1

Reviewer 1 Report

The paper of Z. Diao, X. Hou et al on the fabrication of the hierarchical HZSM-5 membrane for catalytic cracking of n-dodecane is well performed and present interesting performances. However, there is no direct comparison with known membrane/catalyst.

I agree that, sometimes, such comparison with literature data are difficult because the experimental conditions are different and/or the operating conditions could not be exactly reproduced.

Meanwhile, for this particular topics, cracking of n-dodecane, one of the corresponding authors already published the similar experiments but using different template for the synthesis of zeolites (Chemical Engineering Science 135 (2015) 452–460, ref 15). It was also claim that in the latter publication that “However, the mechanical stability of the coating membrane was unsatisfactory” (lines 51-52).

However, when comparing the catalytic cracking activities of zeolite membranes for supercritical n-dodecane under the same conditions (500°C, 4 MPa ,10 mL/min) the deactivation rate look similar. One could also noticed that the conversion are much higher.

It would be interesting for the reader, to compare the different membrane under the same conditions and discuss the results instead of another characterization of another membranes without comparison.

Comparison with classical wall coated zeolites is also strongly encouraged.

Author Response

Point 1: There is no direct comparison with known membrane/catalyst. It would be interesting for the reader, to compare the different membrane under the same conditions and discuss the results instead of another characterization of another membranes without comparison. Comparison with classical wall coated zeolites is also strongly encouraged. 

Response 1: we added the descriptions on comparison of N2 gas permeation over membranes in this paper with that reported by Ji et al (Appl. Catal. A: Gen., 2014, 482, 8-15, ref 33). (L167-171 in revision)  And we found the hierarchical HZSM-5 membrane showed higher gas permeation in comparison to b-oriented bi-layered HZSM-5 membranes, although the loading amount and thickness were both larger for the hierarchical HZSM-5 membrane. This further demonstrated that the mesopores in membrane benefited the diffusivity.

    Furthermore, the catalytic performance over membranes in this paper was also compared with membranes or coatings in literatures (Appl. Catal. A: Gen. 2013, 462–463, 271-277, ref 20; Appl. Catal. A: Gen. 2014, 482, 8-15, ref 33; Appl. Catal. A: Gen. 2017, 533, 90-98, ref 14).  And the descriptions were shown in L250-263 in revision. It was found that hierarchical HZSM-5 membrane in this paper exhibited high catalytic activity and limited extent of secondary reactions due to improved diffusivity.

Point 2: Meanwhile, for this particular topics, cracking of n-dodecane, one of the corresponding authors already published the similar experiments but using different template for the synthesis of zeolites (Chemical Engineering Science 135 (2015) 452–460, ref 15). It was also claim that in the latter publication that “However, the mechanical stability of the coating membrane was unsatisfactory” (lines 51-52).

However, when comparing the catalytic cracking activities of zeolite membranes for supercritical n-dodecane under the same conditions (500°C, 4 MPa, 10 mL/min) the deactivation rate look similar. One could also notice that the conversion are much higher.

Response 2: It was observed by Liu et al. that the mechanical stability of the coating membrane was unsatisfactory and the mass loss of the coatings would reach up to 12.5% and 30% after ultrasonic and thermal shock treatment, respectively. And they found that the catalytic activity and stability of n-dodecane cracking were consistent with the diffusion performance of coatings rather than the mechanical stability (Energ. Fuel., 2011, 26, 1220–1229, ref 16). This implies that the reaction time (30 min) may be not enough for reflecting the influence of mechanical stability on catalytic performance of membrane. Furthermore, the mass loading of coatings in our previous work (Chem. Eng. Sci., 2015, 135, 452–460, ref 15) is about 5.5 mg·cm-2, which is much higher than that of membranes in this paper (1.4 mg·cm-2). And the higher catalytic activity for the former one would be attributed to the higher mass loading.

Reviewer 2 Report

Comments to the authors

The authors report a synthesis of mesoporous HZSM-5 zeolite membranes for cracking of n-dodecane. The mesoporosity, diffusivity and acidity can be tuned through the control over the ratio of TPOAC to TEOS, as well as TMB to TPOAC, which is critical in the catalytic cracking of n-dodecane. The work is interesting and can be considered as a publication in Catalysts. But before publishing it, the authors should address my following concerns.

1. It should be specific surface area instead of special surface area (please check in both abstract and conclusion).

2. The authors are suggested to add the standard diffraction peaks of HZSM-5 and stainless steel tube support in Figure 1 for clear comparisons.

3. Which diffraction peak in Figure 1 do the authors think has decreased intensity versus increased TPOAC/TEOS ratio? The MFI topology peaks look comparable with different TPOAC/TEOS ratios. Also, it is not recommended to conclude the crystallinities evolution based on the absolute peak intensity change. They probably need to look into the relative peak intensity change for accurate trend in crystallinity evolution.  

4. I am wondering what the catalytic efficiency (products yield/specific surface area) is like for these zeolite membranes. Is the ratio of products yield to specific surface area consistent over these zeolite membranes (considering they have identical chemical nature)? 

5. The authors are suggested to make comparisons of their catalytic performances with those reported in the published papers.

Author Response

Point 1: It should be specific surface area instead of special surface area (please check in both abstract and conclusion). 

Response 1: Thanks for your reminding and we corrected linguistic errors, “specific surface area” was used instead of “special surface area”.

Point 2: The authors are suggested to add the standard diffraction peaks of HZSM-5 and stainless steel tube support in Figure 1 for clear comparisons.

Response 2: We added the standard diffraction peaks of HZSM-5 (ICDD PDF No. 44-0003). The X-ray diffraction of stainless steel tube support was measured, which would be relatively clear and intuitive for comparisons between the support and membranes. And all results were presented in Figure 1.

Point 3: Which diffraction peak in Figure 1 do the authors think has decreased intensity versus increased TPOAC/TEOS ratio? The MFI topology peaks look comparable with different TPOAC/TEOS ratios. Also, it is not recommended to conclude the crystallinities evolution based on the absolute peak intensity change. They probably need to look into the relative peak intensity change for accurate trend in crystallinity evolution.

Response 3: we calculated the XRD relative crystallinities (Rc) based on the peak intensity at 2θ=23.1° as compared to that of ZM-0TP, according to the literature (J. Memb. Sci., 2012, 415-416: 57-65). The results were listed in Table 1. And we found the values of Rc decreased with increasing the TPOAC/TEOS ratios, which was in consistent with the results reported by Peng et al. (J. Memb. Sci., 2018, 549: 446-455).

Point 4: I am wondering what the catalytic efficiency (products yield/specific surface area) is like for these zeolite membranes. Is the ratio of products yield to specific surface area consistent over these zeolite membranes (considering they have identical chemical nature)? 

Response 4: we calculated the yields of liquid, aromatics and paraffin products as well as the relevant ratios of yield/specific surface area. It was found that the catalysts with similar acidity (ZM-0.05TP, ZM-0.1TP, ZM-0.1TM and ZM-0.2TM) exhibited the nearly consistent ratios of yield/specific surface area. For instance, the yields of aromatics products/specific surface area were 0.0002 %/(m2·g-1) for all the four samples. The yields of paraffin products/specific surface area were 0.0177, 0.0180, 0.0178 and 0.0189 %/(m2·g-1) for ZM-0.05TP, ZM-0.1TP, ZM-0.1TM and ZM-0.2TM, respectively. And the yields of liquid products/specific surface area were 0.0251, 0.0261, 0.0264 and 0.0294 %/( m2·g-1) for ZM-0.05TP, ZM-0.1TP, ZM-0.1TM and ZM-0.2TM, respectively. Considering the relationship between catalytic performance and diffusion as well as acid properties can be illuminated by mass selectivity of cracking products. Thus, the data of both product yields and ratio values of yield/specific surface area were not listed in our manuscript.

Point 5: The authors are suggested to make comparisons of their catalytic performances with those reported in the published papers. 

Response 5: we added the descriptions on comparison of N2 gas permeation over membranes in this paper with that reported by Ji et al (Appl. Catal. A: Gen., 2014, 482, 8-15, ref 33). (L167-171 in revision)  And we found the hierarchical HZSM-5 membrane showed higher gas permeation in comparison to b-oriented bi-layered HZSM-5 membranes, although the loading amount and thickness were both larger for the hierarchical HZSM-5 membrane. This further demonstrated that the mesopores in membrane benefited the diffusivity.

    Furthermore, the catalytic performance over membranes in this paper was also compared with membranes or coatings in literatures (Appl. Catal. A: Gen. 2013, 462–463, 271-277, ref 20; Appl. Catal. A: Gen. 2014, 482, 8-15, ref 33; Appl. Catal. A: Gen. 2017, 533, 90-98, ref 14).  And the descriptions were shown in L250-263 in revision. It was found that hierarchical HZSM-5 membrane in this paper exhibited high catalytic activity and limited extent of secondary reactions due to improved diffusivity. 
